# Appraising the Sonic Environment: A Conceptual Framework for Perceptual, Computational, and Cognitive Requirements

**DOI:** 10.3390/bs15060797

**Published:** 2025-06-10

**Authors:** Tjeerd C. Andringa

**Affiliations:** SoundAppraisal BV, Kerkstraat 102, 9745 CL Groningen, The Netherlands; tjeerd@soundappraisal.eu

**Keywords:** hearing, sonic experience, soundscape, auditory scene analysis, core cognition, sound appraisal, attention, auditory gist, noise sensitivity, audible safety

## Abstract

This paper provides a conceptual framework for soundscape appraisal as a key outcome of the hearing process. Sound appraisal involves auditory sense-making and produces the soundscape as the perceived and understood acoustic environment. The soundscape exists in the experiential domain and involves meaning-giving. Soundscape research has reached a consensus about the relevance of two experiential dimensions—pleasure and eventfulness—which give rise to four appraisal quadrants: calm, lively/vibrant, chaotic, and boring/monotonous. Requirements for and constraints on the hearing and appraisal processes follow from the demands of living in a complex world, the specific properties of source and transmission physics, and the need for auditory events and streams of single-source information. These lead to several core features and functions of the hearing process, such as prioritizing the auditory channel (loudness), forming auditory streams (audibility, primitive auditory scene analysis), prioritizing auditory streams (audible safety, noise sensitivity), and initial meaning-giving (auditory gist and perceptual layers). Combined, this leads to a model of soundscape appraisal yielding the ISO quadrant structure. Long-term aggregated appraisals lead to a sonic climate that allows for an insightful comparison of different locations. The resulting system needs additional validation and optimization to comply in detail with human appraisal and evaluation.

## 1. Introduction

The central thesis of this paper is that the hearing process continuously appraises the sonic environment, monitors its quality and development, and activates the listening process in response to salient events or streams. The main features of sound appraisal can be mapped on a plane (actually a circumplex) spanned by an unpleasant–pleasant axis and an uneventful–eventful axis, which leads to a quadrant structure of general soundscape appraisals: calm, lively/vibrant, chaotic, and monotonous/boring ([3]; [10]; [16]; [28]; [35]) (see Figures 2 and 3).

Soundscape research ([24], [23]; [56]; [57]) studies how individuals or groups perceive and understand their daily sonic environment. This relatively young discipline has reached a consensus on the two-dimensional appraisal structure that has been validated across various languages and cultures ([3]). This structure is formalized in the ISO 12913-1 ([51]) soundscape questionnaire standard.

Sound appraisal is inherently subjective; different individuals may appraise the same sonic environment differently. However, everyone uses the same basic appraisal dimensions, and individuals broadly agree with each other’s appraisals. Formulated appraisals are crucial for communicating the main qualities of the sonic environment and improving it, particularly in urban settings ([2]; [78]) and care environments for vulnerable individuals ([87]; [60], [61]).

The development of automated soundscape quality monitoring systems benefits from understanding the algorithmic basis of human sound appraisal. This paper aims to provide a first-principle derivation of the perceptual and cognitive processes underlying soundscape appraisal and their computational requirements. This paper also reflects on the author’s career, connecting various research efforts—such as continuity-preserving signal processing, the development of verbal aggression detection systems, soundscape research, and formulating requirements that being alive imposes on cognition—to elucidate the structure of soundscape appraisal. By integrating these diverse areas, this paper offers a comprehensive (but, given editorial constraints, somewhat superficial) explanation of how the hearing process generates an appraisal of the sonic environment.

This narrative focuses more on consistency across a wide range of contributing disciplines than on detailed conformance with experimental results. Contributing domains and topics include neurophysiology, psychophysics, attention research, mood, emotion, stress, restoration, psycholinguistics, speech recognition robustness, soundscape research, and (auditory) phenomenology.

This paper aims to derive, from first principles, a conceptual framework for the perceptual, computational, and cognitive requirements of soundscape appraisal (and hearing). It starts with context and requirements that lead to many constraints. Then, it covers the core features of the appraisal process. Finally, it addresses the implementation of the core function of soundscape appraisal and the characterization of the local sonic environment. The following topics are covered in order:


**Context, requirements, and some definitions**


*Enactive cognition and core cognition*: Enactive cognition (which postulates that all cognition arises from real-world interaction) and core cognition (the hypothesized cognition shared by all of life) ([14]; [7]; [36]) form a suitable framework for understanding hearing and suggest a quadrant structure of mental states. Both frameworks stress the importance of perception–action relations, agency, and sense-making. Soundscape appraisal is a sense-making process ([90]).*Properties of sound*: The physics of sound production and propagation ([6]; [42]) makes it ideal to monitor events in the proximal environment omnidirectionally, and it points to algorithmic approaches to auditory stream segregation.*Audition = hearing + listening*: The pre-attentive hearing process produces soundscape appraisal, including saliency indicators that activate and orient the listening process for attentive, detailed, and goal-oriented analysis ([6]).*Pre-attentive processing*: Hearing is intimately interwoven with other subcortical functions, such as mustering and reorienting attentional resources ([48]), arousal, and estimating threat and safety ([76]). It contributes to a mood-level appraisal of the (sonic) environment known as the core affect ([79]).*Source physics*: The structures that physical limitations of sound sources impose on their sounds allow for robust evidence estimation ([5]), source classification, and perceptual constancy despite transmission effects. Source physics and other environmental regularities underlie predictive coding ([20]). Source physics also conveys whether the sound production system is free to produce a normal variant of its source or is under stress, distorting the sound away from the norm ([91]).*From sound to sounds*: Recognizing individual sources requires the transition from undifferentiated *sound* to individual *sounds* ([6]) that can be attended to and further divided or regrouped into source-specific auditory streams. This categorization process imposes many severe constraints on auditory processing.*Primitive auditory scene analysis*: This process creates auditory events and streams of information about individual (and concurrent) sound-producing events ([25]). It relies heavily on tracking whatever structure source physics imposes on the produced signal. Violated expectations about this development are typically assumed to be caused by another (new) source ([21]). This is a manifestation of the old-plus-new heuristic ([25]) that reoccurs throughout this paper.*Noise*: Noise results from initial meaning-giving on a pleasant/unpleasant axis, and it is a valence-related form of sense-making ([10]).


**Core features of the hearing process**


9.*Loudness*: Loudness weighs the importance of the auditory channel compared to other senses and priorities ([10]).10.*Audibility*: This measures how far (in dB) a signal component is above the local background and how easily the component musters the cognitive resources necessary for a meaningful analysis ([4]).11.*Audible safety*: Estimating audible safety is a reason d’être of audition in determining whether one is free to self-select activities or be forced to be vigilant ([10]; [87]; [60], [61]). This leads again to a quadrant structure.12.*HiFi and LoFi soundscapes*: This distinction, which originated from soundscape pioneer Shafer ([81]), relates directly to how easily an auditory scene can be analyzed and safety estimated due to the presence of sufficient and easily estimated signal components.13.*Processing up to the level of estimated irrelevance*: The hearing process must apply a strategy of processing sensory input up to the level of estimated irrelevance (in terms of audible safety and valence). Auditory streams need to be processed further until their irrelevance can be estimated. When the hearing process does not estimate irrelevance, the listening process will be tasked with additional analysis.14.*Noise sensitivity*: This prioritizes the auditory channel with respect to the other sensory channels. Multiple forms of noise sensitivity exist: one for louder sounds far away and the other for subtle sounds close by ([54]). A third form results from developing source-specific detection expertise, especially in noise-annoyed individuals.15.*Auditory gist*: The hearing process does not produce a full analysis; instead, it produces a basic categorization called the auditory gist ([48]), suitable for activating situationally appropriate behaviors, kick-starting (bootstrapping) the listening processes, and holistically evaluating the auditory environment.16.*Four types of attention*: The hearing process facilitates (and impedes) four different forms of attention: fascination ([58]), free-roaming directed attention (flow) ([34]), distracted attention ([67]; [80]), and undirected attention ([38]; [77]).17.*Perceptual layers*: Different sound sources’ temporal properties (an aspect of source physics) drive the activation of these forms of attention, leading to perceptual layers (by applying the old-plus-new heuristic) and appraisals associated with individual sources and the sonic environment as a whole.


**Sound appraisal implementation**


18.*Soundscape and sound appraisal*: A more precise definition of soundscape is used to define *sound appraisal* as describing “the character of the acoustic environment” ([1]).19.*Properties of the perceptual layers*: We combine the time scale of vocalization with the requirements of estimating audible safety to derive a link between signal content and sound appraisal.20.*Sound appraisal estimation* combines evidence indicative of each of the appraisal quadrants on a per-second (moment-to-moment) basis.21.*Sonic climate estimation*: Sound appraisals are aggregated across time to provide a simple and informative (quadrant-based) graphical depiction of the characteristic of the auditory environment that is generally stable over time.22.*Measuring annoyance likelihood*: This addresses an informal, but useful, result derived from multiple interactions with community-sound annoyance issues.

## 2. Context and Requirements

This section outlines the starting point for the formulation of the perceptual, computational, and cognitive requirements of sound appraisal. It comprises several theoretical considerations. The hearing process can be understood in the wider context of the cognitive processes of survival and flourishing that we approach from the theoretical standpoint of enactive cognition ([14]; [70]) and core cognition, which provides a suitable language and a quadrant structure from which we can understand much of the phenomenology of hearing and soundscape. The hearing process is embedded within other subcortical processes and interacts with and contributes to them. Another important starting point is the source and transmission physics that shape and degrade the auditory input because these offer opportunities and impose limitations on signal processing and signal representation.

### 2.1. Enactive Cognition and Core Cognition

This paper starts from the meta-theoretical position of enactive cognition ([14]; [69]; [70]) and core cognition ([13], [14]; [7]; [90]; [36]) because it provides a clear and consistent framework for connecting cognitive functions, like the hearing process, to the demands of survival and flourishing. Enactive cognition can be summarized succinctly as “being by doing” ([41]), which, in the case of the hearing process, boils down to the question “What do I need to know to decide how and when to act?” Acting refers to overt and mental activities that eventually produce situationally appropriate overt activities.

Core cognition uses the axes, as depicted in Figure 1, to hypothesize the basic cognition common to all living beings ([7]). The starting point is that all living agents maximize their probabilistic distance to death. Agents do this by *reactively* adapting to and *proactively* asserting themselves in the environment using its affordances while satisfying the viability-preserving demands it imposes. This allows them to cope with environmental challenges and to co-create an environment more likely to sustain future need satisfaction. We separate two essentially different cognitive processes: *co-creation* aims to avoid pressing problems from occurring, and *coping* aims to solve pressing problems as quickly and effectively as possible. Effective co-creation allows the agent to learn, care for, or improve the state of self and the environment, increasing its probabilistic distance to death. Effective coping allows the agent to negotiate and solve problematic situations. Ideally, co-creation is the behavioral default, with coping as a temporary fallback.

The diagonal axes refer to the associated appraisal of the environment and attentional states suitable to match its possibilities and challenges. The diagonal to the upper right denotes an affordance axis, with an environmental appraisal on the left side as deficient and a rich or enriching one on the right side. The other diagonal refers to viability-impacting constraints on behavior. Few or no constraints on the lower right lead to an environment appraised as safe. Many severe constraints on the left lead to a dangerous situation where suboptimal action selection and action execution lead to severe (even lethal) consequences.

These two sets of (dependent) axes lead to a natural separation in quadrants ([7]; [10]; [90]; [36]) in which cognitive, here attentional, resources are coupled to the appraisal of the environment ([10]; [15], [16]; [17]). This also connects directly ([7]; [90]) to the concept of the core affect ([89]; [62], [62]; [79]), which refers to how moods are structured. The core affect, in contrast to emotions, is continually available for introspection ([79]). We develop this later in the context of the different manifestations of attention in soundscape appraisal.

*Moods* are conceptualized as a relationship towards the world perceived as a whole, while *emotions* are a relationship towards *something* in the world ([14]; [87]; [90]). Because hearing processes involve more holistic and superficial meaning-giving, they connect to moods. Listening connects to emotions. The mood-level quadrants underline the soundscape structure that the ISO 12913-1 soundscape questionnaire refers to.

### 2.2. Properties of Sound

As we argued ([6]), compared to light, sound travels across less distance, which limits sonic information to the proximal environment. And because auditory information is spectrally oriented, spectral information from all directions is pooled, which makes auditory sensitivity omnidirectional (vision is highly directional because only a 3° section can be analyzed in detail). This omnidirectionality, in combination with sounds being informative of material interactions ([42]) in the proximal environment, implies that audition is highly suitable for proximal omnidirectional monitoring. “Proximal” has two extremes: subtle sounds close by and loud sounds from afar ([54]). Both domains are important. Near sounds indicate states (e.g., normalcy) and developments (e.g., a change of character) or disturbances in the immediate environment, and they also offer feedback on one’s movements (e.g., one’s footsteps). Sounds from afar, especially looming sounds, may indicate approaching dangers.

These properties ensure that audition is highly suitable for omnidirectional monitoring of the proximal environment, especially for potentially viability-impacting situations. This not only allows one to respond to dangers but also to estimate whether a situation is safe and conducive for self-selected behavior (e.g., to learn or for self-care).

### 2.3. Audition = Hearing + Listening

In [6] ([6]), we used common sense (dictionary) definitions to define audition (noun) as “*the capacity for, or act of sound-based processing in which the existence of something or someone becomes mentally available (in the case of awareness), this availability can be used in a reasoning process to discover the consequences of what has been perceived (in the case of consciousness)*”. This definition separates two processes. This paper makes these more precise: “*The preattentive hearing process produces a soundscape appraisal—including saliency indicators that activate and orient the listening process—for attentive, detailed, and goal-oriented analysis*”.[note 1]

### 2.4. Pre-Attentive Processing

The nuclei and fiber pathways of the brainstem play a central role in various interrelated sensory and motor functions. These include mediating and controlling attention, arousal (and sleep), heart rate, breathing, balance, initiating and coordinating the eye, jaw, tongue, and head movements, and somesthetic, gustatory, auditory, and visual perception. They balance the sympathetic (e.g., fight or flight, activities) and parasympathetic nervous (e.g., restoration, self-care, immune functions) systems to maintain homeostasis during different states of arousal or relaxation ([46]; [55]; [75], [76]).

Unexpected, context-inappropriate, or simply loud sounds can lead to a brainstem response—a form of bottom–up or stimulus-driven attention—in which ongoing neocortical processes are interfered with to address sensations that the brainstem considers worthy of attentive processing ([20]; [22], [21]; [94]). Attentive—top–down or knowledge-driven—processing requires resources that become available through arousal. At the same time, the sounds may lead to the disruption of ongoing tasks through an automated and rapid reorientation towards the sound that may also involve involuntary eye movements. All of these changes are intertwined with the hearing process. In actuality, the hearing process might not exist as a truly separable process from the rest of the subcortical functions.

Porges introduced the term neurception as “a subconscious system for detecting threat and safety” ([75]). He concludes that “when humans feel safe, their nervous systems support the homeostatic functions of health, growth, and restoration, while they simultaneously become accessible to others without feeling or expressing threat and vulnerability” ([76]). To implement core cognition, we assume that this pre-attentive system produces an analysis (appraisal) of the state of the environment in terms of the core cognition quadrants and the associated activation of cognitive (attentional) resources. This results in a mood-level, core affect ([10]; [79]) underlying four broad soundscape evaluations (see below).

### 2.5. Source Physics

The physical nature of auditory knowledge is important ([5], [6]). Gaver argues ([42], [43]) that if suitably processed, sonic input leads to complex but structured patterns that are informative of the sources that produced the sounds. Knowledge-driven (top–down) attentional processes should detect and capture these predictable structures ([20]) as predictive coding ([20]; [40]; [47]). An important problem is that signals reflecting ideal source physics rarely stimulate the cochlea since a perfect source signal degrades due to transmission and mixing with other sounds. The details of this “degradation” are a rich source of information that bats, dolphins, and many blind people have learned to use effectively ([85]).

The strong relation between physics and knowledge ([6]) can be demonstrated by a thought experiment, adapted from [5] ([5]) as the following question: “Which sound source cannot be recognized?”. We might perform an experiment by setting the sound source behind an opaque screen. First, you hear a sound and say “A violin”. “That is correct”, we say. Then you hear another sound. You again report a violin. “Wrong”, we say. But you heard the violin. We remove the screen, and you see both a violin player and a HiFi set with very good loudspeakers. The loudspeakers are sound sources, and they tricked you the second time.

This experiment might not seem particularly informative because this is exactly what loudspeakers are used for. However, the point is that the violin will always produce ‘the sound of a violin’; it is the only sound that its physics allows it to produce and that our auditory system allows us to interpret. The same holds for all other “normal” sound sources. In contrast, the (ideal) HiFi set can reproduce any sound. It has no audible physical limitations and, consequently, has no characteristic sound. It will always be interpreted as another sound source if the listener is naïve about the true origin of the sound. And even then, it is effortless to interpret a HiFi set as the sound sources it reproduces.

In practice, the hearing process should estimate evidence of source physics from a noisy transmission channel with evidence of multiple overlapping sources degraded by transmission effects. Importantly, while the source signal can be severely degraded, we rarely notice this: we become aware of an idealized physical interpretation of the signal, not the signal itself (perceptual constancy ([32])). This entails that the voice of a friend sounds similar and recognizable despite loudness differences and severe signal degradation from transmission effects. And this holds generally.

Vocalizations and other sounds produced under stress change predictively. Very loud vocalizations tend to change from sinusoidal towards block waves with stronger high-frequency contributions. Stress on the muscles that produce the sound leads to less fluent transitions and a more “robotic” quality. Stress-induced vocal fold stiffness leads to a higher pitch. These three stress-induced changes co-occur in aggressive and panicked voices and have been used to detect verbal aggression ([91]; [59]) and are manifestations of how changes in source physics impact the produced vocalization.

### 2.6. From Sound to Sound

Recognizing sounds entails separating the undifferentiated ‘sound’ that reaches the ears into ‘sounds’: separated *streams* of acoustic information stemming from one or more physical sources. This process is known as auditory scene analysis ([25]). We refer to this as “Audition: from sound to sounds” ([6]). Continuity-preserving signal processing ([5]) was developed to track source physics without introducing non-physical discontinuities that complicate estimating and tracking source physics. Hence, it allows us to isolate physically coherent information: *signal components* described by an onset, some form of continuous development, and an offset. Temporal coherence ([20]; [82]) within and across signal components leads, in combination with the old-plus-new heuristic ([25]), to the formation of auditory events. On- and offsets can result from the emergence above competing sounds or the physics of starting or stopping the sound-producing event. The formation of differentiated sounds is directly related to categorical perception ([19]; [49]) (and perceptual constancy ([32])).

### 2.7. Primitive Auditory Scene Analysis

The isolation of signal components from an undifferentiated sound is called *primitive auditory scene analysis* ([25]). Primitive ASA transforms the undifferentiated input sound into discrete “events”. Physical sources produce multiple correlated signal components. The correlations stem from a common location, common on- or offset, common amplitude or frequency modulation (AM, FM), or, like in the cases of speech and music, multiple physical sources exhibiting a higher order (linguistically and culturally) structured—hence predictable—pattern. Normally, individual sound sources are uncorrelated, producing sounds with strong internal correlations among their signal components and weak (occasional or temporary) external correlations with the signal components of other sound sources.

Highly correlated signal components, like the onset, harmonics, and offset of a single musical instrument, should not be separated and perceived as a single entity, such as an instrument’s sound (with qualities like timbre and pitch). This holds on different levels. For example, the coordinated sounds of an orchestra can be perceived as the orchestra, as multiple groups of instruments playing separate melodies, and, in many cases, as individual instruments. This all depends on how precisely we can focus our attention on the signal constituents, which in turn depends on the degree to which we can use directed attention, which also depends on how well we can predict the development of source properties. The more we can direct our attention to detail, the more detail we perceive, and the less we perceive the details of the unattended part. This form of prediction-guided directed attention is a characteristic feature of listening and “*schema-based auditory scene analysis*” ([25]).

The hearing process is limited to a more superficial, but still highly complex, initial separation of sounds. This separation can occur in several auditory streams representing and tracking individual sources like a voice, a bird sound, or the sound of a passing car. It can also yield difficult-to-separate ensembles of sources such as the babble sound of a cocktail party, the early morning bird choir, or distant traffic. In our modeling, we use the characteristic temporal development of sources to separate perceptual layers (using the old-plus-new heuristic), implementing primitive auditory scene analysis. It seems that we do not need schema-based or knowledge-driven ASA to explain the main features of sound appraisal. Hence, we conclude that sound appraisal is an outcome of the hearing process.

### 2.8. Noise

Music is generally assumed to be pleasant or at least interesting; in contrast, noise is any sound considered unpleasant by an individual, and it interferes with the reception of wanted sounds ([10]; [93]). Pleasant/unpleasant or wanted/unwanted decisions are forms of basic appraisal and sense-making that should have a place in the hearing process and, in fact, fit with basic avoidance and approach decisions and, in general, valence.

Figure 2 gives an overview of the hearing process’s most important appraisals, especially the distinction into four “experience quadrants” and their most important properties. This division into quadrants is for communication convenience; the underlying structure is circular ([90]) and not categorical.

## 3. Core Features of the Hearing Process

Where the previous section provided some perceptual, computational, and cognitive starting points of soundscape appraisal, this section describes its main features and capabilities. It addresses the roles of loudness and audibility, positive safety indicators, initial meaning-giving for prioritizing signal properties, and the activation of attentional resources. The section ends with the observation that the temporal dynamics of sources lead to the natural separation of the sonic environment that the hearing uses to analyze, appraise, and make sense of the sonic environment.

Although pitch, timbre, roughness, and sharpness are clearly appraisals and outcomes of primitive auditory scene analysis, we ignore these in the current paper since the practice of noise abatement and soundscape focuses (too heavily) on loudness. In addition, pitch, timbre, roughness, and sharpness involve additional complexity extending beyond the scope of the current paper.

### 3.1. Loudness

Loudness, like what a decibel meter measures, is a measure of the total acoustic energy stimulating the hair cells of the cochlea and the auditory nerve. Without further processing or situational knowledge, it says little about what produces it or how it will be experienced. However, within the wider context of (pre-attentive) cognition, it can be used to “weigh” the importance of the auditory channel compared to the other senses ([18]; [93]) and priorities. Loudness is an auditory equivalent of pain for touch: a sensation that draws attention to a sensory modality and recruits cognitive resources.

Interpreted this way, it makes sense that loudness measures (e.g., L_den_, L_A,eq_) have limited applicability in protecting and improving the sonic environment. Loudness-based measures and regulations are convenient and important for preventing excesses but are of very limited use in optimizing the quality of the sonic environment ([12]). The soundscape measures presented later in this paper are far superior because they pertain to meaning-giving.

Loudness can also be assigned to individual signal components, but only after (at least) primitive auditory scene analysis (or otherwise ensuring that a single source dominates). The loudness of individual signal components or streams can be combined with source knowledge to determine distance, movement, or source power. However, this uses source properties, situational/contextual knowledge, directed attention, and attentive listening, and should be assigned to the listening process. That red cars appear slightly louder than blue or light-green cars ([66]) results from this downstream loudness perception. Similarly, this underlies the perceptual constancy of voices at varying distances.

### 3.2. Audibility

*Audibility* indicates how easy it is for primitive auditory scene analysis to isolate a signal component from the whole. Highly audible signal components are estimated easily, reliably, and quickly, therefore easily capturing attention. For short sounds (bird calls, speech, car passages), audibility corresponds to how far (in dB) the sound is above a slower-changing background. Stationary sounds (a distant highway, continuous rain) constitute a background and are audible but not particularly salient. Background sounds become more audible and salient when they have tonal components (e.g., an aircon needing maintenance) or amplitude modulations (e.g., wind gusts). Since audibility is integral to auditory scene analysis, it is more useful than loudness.

The inverse of audibility is listening effort. Studies indicate that as audibility decreases, the degraded signal is less effective in recruiting the (memory) resources required to process it ([4]; [64]; [65]). A source like a target speaker can be recognized if sufficient resources can be recruited in real-time (either via prediction or stimulus-driven). If not, prediction quality degrades quickly and leads to a matching performance loss. This plays a role in the cocktail party effect ([26]; [29]; [50]).

### 3.3. Audible Safety

When highly sound-annoyed individuals are asked which sounds they actually like, they come up with an eclectic range of examples: “The sound of my husband reading a book”, “The sounds of my kids playing in the other room”, “The sound of a cow eating”, “The sound of my neighbor cleaning her house”, “Birds in the park”, “A busy café”, and “Relaxed chickens”. These sounds have very few common acoustic properties. What they do have in common is that all sounds refer to relaxed individuals making sounds, indicating that they feel relaxed and safe. This leads to the concepts of *audible safety* and *positive indicators of safety* ([10]; [87]; [90]; [61]). For example, [39] ([39]) concluded “that if sites were perceived as species-rich, containing natural sounds like birdsong, natural rather than artificial, and safe, they were perceived as more restorative, resulting in improved wellbeing”.

Estimating audible safety rests on the unique physical properties of the sound modality (omnidirectional indications of change or approach) and, hence, on the *reason d’être* of audition. *Positive* safety indicators help determine whether one is free to self-select activities. Their absence forces one to be vigilant. And *indicators of unsafety* force a response. As we argued ([90]), audible safety allows us to interpret the core cognition quadrants in terms of basic soundscape categories: calm, vibrant/lively, chaotic, and boring/monotonous. Specifically, a calm environment affords ample indications of safety that allow us to restore our resources, care for ourselves, and tend to the environment; a lively environment is a stimulating and safe place that allows us to learn and play; a boring environment misses indications of safety, which does not afford a sense of safety or control; and a chaotic environment contains clear indications of insecurity or danger and forces us to retain or regain control. Figure 3 shows this graphically. 

### 3.4. HiFi and LoFi Sounds

Audible safety estimation can be coupled to ease of information processing and the concept of a HiFi and LoFi environment ([81]). A HiFi soundscape has few overlapping sources and well-audible signal components that primitive auditory scene analysis processes can easily estimate and analyze. Hence, these can recruit the top–down knowledge necessary to categorize and attribute meaning. This allows for processing at a higher event rate compared to a LoFi environment. In a LoFi environment, the signal often changes before it is fully processed, contributing to a lack of certainty about the state of the environment and low audible safety.

Animal-dominated sonic environments are typically organized such that smaller species with more numerous individuals produce short, high-frequency sounds ([74]; [86]). In contrast, less numerous larger individuals produce fewer, longer, and lower sounds. This leads to a layered spectral structure, a distant sonic horizon, and a high signal-to-noise ratio (i.e., a high foreground-to-background ratio and high audibility), allowing source information to be minimally distorted and allowing relevant information about the environment to be estimated at a distance. It also allows for a high rate of events that can be separated and tracked.

Alternatively, LoFi soundscapes are associated with an industrial, mechanized world with loud broadband noises (such as traffic sounds) that mask natural sounds, can be associated with signal distortions, and lead to a narrower sonic horizon. The masking and distortions complexify the analysis and make it more difficult to identify and track relevant events. As such, a HiFi (often natural) soundscape is pleasant because it is favorable for survival since it is easier to process and informs about a wider environment. Conversely, LoFi is unpleasant and unfavorable due to the masking of signal information, distortions, and a reduction in the sonic horizon ([90]). Hence, it is not so much that LoFi environments are unsafe or dangerous, but rather they complicate or block the estimation of *positive* indicators of audible safety, preventing relaxation and hindering behavioral self-selection.

### 3.5. Processing up to the Level of Estimated Irrelevance

Processing up to the level of estimated irrelevance is an important feature of all perception, but especially for the hearing process due to its essential temporal characteristics. The most important function of the hearing process is to ignore all sounds that are not currently a priority. This requires a quite reliable level of processing and classification of the whole input up to the point of estimated irrelevance. Hence, the role of primitive scene analysis is to separate the undifferentiated sound into differentiated sounds, perform a basic meaning-giving analysis of all the sounds on a valance dimension, and prioritize relevant auditory streams for the listing process. The words in blue in Figure 2 are examples of this basic meaning-giving and sense-making.

From an evolutionary perspective, estimating audible safety must be fast and “*better safe than sorry*”. Pre-attentive cognition must keep the animal safe in a complex world, and it can only do that with basic learning processes such as conditioned responses and habituation. For example, the hearing process might learn that a certain stimulus, frequent or prolonged, co-occurs with emotions or required responses and can tune itself to become more sensitive (e.g., to your name or your baby’s sounds). This *conditioned response* makes the stimulus more salient without requiring conscious control. Similarly, the hearing process can learn that strong or initially unexpected stimuli do not warrant a strong response (*habituation*).

### 3.6. Noise Sensitivity

Some features of this prioritization manifest in the concept of noise sensitivity. Noise sensitivity entails a sensitivity to—entailing a prioritization of—unwanted sounds, typically sounds that the listener associates with the left side of Figure 1, Figure 2 and Figure 3. Highly noise-sensitive individuals are much more likely to be disturbed by sounds they do not want to listen to (i.e., attend consciously). This can be interpreted as noise-sensitive individuals living in a world they deem less safe and manageable. Hence, their arousal is more likely to be high at the cost of self-care, which is activated in audibly safe low arousal states. Individuals with high noise sensitivity are more likely to experience annoyance and negative emotions such as depression, anxiety, anger, tension, inferiority ([71]), and, in general, low mental health ([84]). High noise sensitivity may, therefore, indicate a coping trap and a high score on the psychopathology dimension p ([63]; [83]).

[54] ([54]) describes two forms of noise sensitivity: one for louder sounds far away and a second for subtle sounds close by. The first form is likely associated with loudness and prioritizes the hearing process’s outcome for conscious processing. The other form prioritizes subtle qualitative changes or intrusions in the local environment via a process known as auditory gist.

A third form of noise sensitivity likely occurs when individuals have learned to associate a negative valence evaluation with certain source-specific signal components that make them more sensitive to them than others. This can even lead to developing perceptual expertise in detecting the sounds one does not want to hear (and can barely ignore and hence is forced to listen to consciously) ([9]).

### 3.7. Auditory Gist

The hearing process produces an initial meaningful evaluation called the *auditory gist* to kick-start (bootstrap) and guide the listening processes. Gist was first described for vision ([45], [44]; [68]) and provides a tentative, meaningful evaluation of the affordance content of a visual scene. Visual gist seems more associated with behavioral options than basic classification into objects. In flashed visual scenes ([45]), behavior-informing properties like *concealment* (“I can hide here”), *mean depth* (“The space is vast (or confined”), *naturalness* (or urban), *navigability* (“I can(not) find my way here”), *openness* (“I can(not) see the horizon”), *temperature* (“It is a hot/cold place”), and *transience* (“I expect this scene to be moving/stationary”) are activated. These action-related properties were, on average, estimated within 34 ms of exposure, while basic-level categorizations like *desert, field, forest, lake, mountain, ocean,* or *river* required, on average, 50 ms of exposure (50% longer).

Harding et al. proposed ([48]) *auditory gist*. In particular, they found auditory (and visual) domain evidence that (1) only the gist of the scene or object is initially processed; (2) processing of the gist is rapid and superficial; (3) the focus of attention is deployed according to prior knowledge and gist perception; (4) conscious detailed analysis is possible on the part of the scene within the focus of attention; and (5) only limited processing of the unattended parts of the scene occurs.

This substantiated the conclusion that audition = hearing + listening. “*Hearing detects the existence and general character of the environment and its main and salient sources. Combined with task demands, this allows the pre-activation of knowledge about expected sources and their properties. Consecutive listening phases, in which the task-relevant subsets of the signal are analyzed, allow the level of detail required by task- and system demands*” ([6]).

This conclusion remains vague about what is meant by the “*general character of the environment and its main and salient sources*” ([6]). This paper proposes that the soundscape descriptors in blue in Figure 2 (derived from ([16])) are typical examples that can be estimated from a superficial, holistic analysis. Many of the concepts around the circle are opposites of some kind: eventful–uneventful, calm–chaotic, beautiful–ugly, interesting–boring, and rural–urban. After primary auditory scene analysis, these can all be estimated from a superficial and holistic evaluation of the ensemble of signal components and auditory streams. They are all forms of meaning-giving and sense-making ([33]; [52]; [53]).

### 3.8. Four Types of Attention

Earlier papers ([10]; [90]) noticed the relationship between the appraisal of the auditory environment and attentional states, and Figure 1 and Figure 2 referred to four different forms of attention. It is now possible to describe these attentional states in relation to hearing and listening.

*Fascination* occurs with safe and inherently attractive stimuli. People can lose themselves in looking at the leaves of trees or a beautiful landscape. Beautiful music works are similar; you can focus your attention on the music effortlessly, and while you do so, you also relax. Fascinating stimuli effortlessly guide your attention, and because a fascinating stimulus is inherently attractive, you like the guidance. Fascination helps us to recover from periods of effortful directed attention (Attention Restoration Theory ([58]; [72], [73]; [78]; [92])).

*Free-roaming focused attention* occurs when you decide, with little effort, where to focus your attention and what activities to perform, for example, reading a book or having a conversation. Many other attractive activities may be possible, but you can easily focus on and immerse yourself in the chosen activity. If that happens for a prolonged period, it is called *flow* ([34]). This form of attention is not particularly tiring. Like fascination, this form of attention is activated in HiFi environments.

*Distracted attention* can be defined as a situation when you aim to focus on a task, but you are distracted by, for example, disturbing noises that prevent you from concentrating or force you to switch between a self-selected task and attending to a distractor. This requires a higher level of arousal ([67]) and is an inefficient use of time and energy. Not only because you focus on an unwanted stimulus that likely elicits negative emotions, but also because it requires considerable switching time to return to your task ([80]). This is (rightly) perceived as highly disturbing, tiring, and annoying ([9]).

Finally, there is a form that we could call *unaimable* or *undirectable attention* associated with boredom ([38]; [77]). This occurs when the environment offers nothing useful, positive, or interesting, for example, when waiting for transportation at an unfamiliar, deserted, and boring place. Nothing is interesting, and no positive safety indicators exist, so you do not feel at ease and remain alert. Boredom “is consistently related to negative effects, task-unrelated thought, over-estimation of elapsed time, reduced agency, and over- and under-stimulation” ([77]). This is not a relaxing form of attention.

### 3.9. Perceptual Layers

Earlier, we stated that the examples of positive indicators of audible safety share few acoustic properties. They share, however, one very important feature: *they were produced by living individuals and developed on the time scales of animal behavior*. They had a time constant of less than a second, and the associated sound events (syllabi, calls, vocalizations, steps, et cetera) typically developed within 1 s from start to end. Vocalizations generally indicate audible safety, but not for vocalizations under stress, like pain, aggression, or panic, which raise pitch and spectral tilt.

It took a while to realize that most other relevant sounds developed on considerably longer time scales. Passing cars typically last 5 to 30 s, trains and aircraft a bit longer, and lawnmowers or leaf-blowers multiple minutes. The city background develops over the day with little difference within an hour.

This led us to propose that (1) the hearing system (in particular, the primitive auditory scene analysis) uses time constants to separate the sonic environment into “perceptual layers” (implementing the old-plus-new heuristic) and (2) that different layers have predictable subjective effects, lead to qualitatively different appraisals and lead to a prototypical quadrant structure as depicted in Figure 4. This will be developed further in the next section.

## 4. Sound Appraisal Implementation

This section shows that the conceptual framework described above can be applied in automated sound appraisal technology, and it provides some examples of applications in soundscape projects. It starts with the technical conceptualization of soundscape.

### 4.1. Soundscape and Sound Appraisal

While discussing the hearing process, the soundscape was not yet defined with sufficient precision. The ISO 12913-1 definition says that “*Soundscape is the acoustic environment as perceived or experienced and/or understood by a person or people, in context*”. This human-centered definition of soundscape comprises both the hearing and listening process and the resulting meaning-giving. Somewhat problematic is that on an individual level, perceiving, experiencing, and understanding are qualitatively different to on the group level because the group level misses the qualia ([27]; [37]) and, by necessity, produces explicit knowledge, typically a shared narrative.

[1] ([1]) recently proposed a variant definition of the soundscape *as* “*the acoustic environment of a place, as perceived, experienced, and/or understood, whose character is the result of the action and interaction of acoustic, nonacoustic, and contextual factors*”. This definition does not specify whether it is experienced subjectively (by an individual and with qualia) or objectively (by a group as a descriptive narrative), and it is also not human-centered. Usefully, this definition describes a result: “*the character of the acoustic environment*”.

The process that leads to this result can be referred to as *sound appraisal*, using the normal dictionary definition of appraisal as “*to evaluate or estimate the nature, ability, or quality of something*”. If individuals or groups share their experiences and evaluations, sound appraisal results in explicit knowledge about the sonic environment. A sound appraisal system should approach these group-level descriptions of the characteristic of the acoustic environment in terms of the concepts in Figure 2.

Note that, defined this way, the soundscape is indeed an outcome of the hearing process (optionally enriched by attentive listening). This outcome can be instantaneous (referring to the here and now) or aggregated over time (and even aggregated over multiple locations within an area).

### 4.2. Properties of the Perceptual Layers

The section on subcortical processing concluded that the hearing process should produce an analysis of the state of the environment in terms of the core cognition quadrants. The associated activation of cognitive (attentional) resources resulted in “attitudes towards the world”—mood words and general world descriptors (see Figure 2)—associated with four broad soundscape evaluations. The features of the perceptual layers facilitate these meaningful analyses. Four perceptual layers were defined as an initial choice, separated by three time constants: 1 s, 1 min, and 1 h. This was the minimal number of layers required to map to all four quadrants.

Summed up, these layers yield the original input sound as a spectrogram or, in our case, a cochleogram because we compute it with a cochlea model ([5]). We derive the first layer by subtracting components that change considerably in 1 s. Next, we subtract components that change considerably in one minute and then subtract components that change considerably in one hour. These three subtracted layers form the second, minute, and hour foreground layers. This leaves a background that changes little in one hour: the hour background. The sum of the hour background and the three foregrounds yields the original cochleogram. Figure 5 shows this process as a cartoon of the old-plus-new heuristic.

The three foreground levels represent audibility in dB above the next (slower) level. They indicate how much energy the layer contributed to the whole.

*Second foreground layer*: This perceptual layer contains mainly human and animal vocalizations and indicates audible safety through relaxed vocalizations. Small animals like birds and insects generally produce high frequencies (>2300 Hz) and contribute to a calm and relaxing evaluation, while human vocalizations, especially in a social context, contribute to a lively/vibrant evaluation. Much of music is also captured by this time constant. Indications of sound production under stress (like sharp onset, a flatter spectrogram, and high pitch for the type of sound) may be indicative of a lack of safety and may contribute to a chaotic or unpleasant appraisal.*Minute foreground layer*: This contains sounds lasting longer than 1 s and developing faster than in 60 s. These sounds, albeit much slower, are still interpreted as changing and developing. This layer contains most passing traffic (cars, scooters, aircraft) and other events that (such as gusts of wind) typically have a broadband character and can mask or complicate the estimation of the audible safety indicators in the second foreground. These sounds are typically interpreted as distractors and lead to a chaotic interpretation.*Hour foreground layer*: This layer contains sounds that last longer than a minute and develop faster than in 1 h. These sounds have a more stationary character. This layer contains slower passages like a high-passing aircraft or the passage of a boat. The sounds of longer mechanical sounds, such as farmers working in their fields or the sound of a stationary truck, are typical. Weather events like rain showers are also likely to occur in this layer. This layer is the transition to events that appear stationary and generally contribute to a monotonous or impoverished evaluation.*Hour background layer*: This background layer contains sounds that remain constant for an hour or more and typically form a “stationary” daily (e.g., city) background. Often, this layer represents rather little energy because of silences between individual sounds. Without these silences, for example, near a busy highway, the hour background can be rather energetic, hence contributing to a monotonous or impoverished evaluation because local details in the second foreground are masked.

### 4.3. Sound Appraisal Estimation

For each second, the soundscape quadrant is determined as a weighted vector sum of the contribution of the four layers to the main soundscape dimensions, *pleasurableness* and *eventfulness,* that specify a 2D point within a circle. The hour foreground is the only layer whose contribution is determined by (approximate) sound levels in dB(A); the other layers use audibility in dB above the previous layer. All contributions scale linearly from 0 to 1 within a value range. Different frequency ranges may contribute to different quadrants. Table 1 gives an example of the values used in Figure 6.

These values lead to very useful and easy-to-interpret results, which comply with experiments in which we asked participants to track the pleasantness and eventfulness of a range of recordings with a joystick ([88]). They also comply broadly with the body of soundscape appraisal experiments that led to the ISO 12913-1 standard. We cannot yet claim that our estimation is close to that of humans in normal soundscape conditions, but we expect a fair to good general agreement (and considerably better after tweaking).

### 4.4. Sonic Climate Estimation

The sonic climate is the long-term aggregate of soundscape evaluations as a 2D histogram. Ideally, it approaches the effort of the average outcome of human volunteers who produce a second-by-second response to the ISO standard questionnaire. Figure 7 shows what results from binning all pleasantness and eventfulness combinations in a 2D histogram (here 21 × 21 for all seconds of a day) on axes spanning [−1, 1]. Figure 7 shows indications for representing, color-coding, and interpreting a sonic climate histogram. The color coding scales with the fraction of the data per bin. Currently, no claims are made about the correspondence of the automatic and human appraisal estimation other than that the resulting 2D histogram roughly corresponds to expectations and is informative even for laymen.

These sonic climate histograms are surprisingly stable over time. Weekly differences in the use of the space or seasonal influences in bird calls influence the higher percentile contours, but similar conditions lead to similar histograms that are interpretable with little training. A comparison of daily histograms can easily identify deviations from the norm.

The sonic climate can be used for a variety of applications, for example, to describe the local soundscape and to determine whether it matches its intended purpose. The six-hourly examples in Figure 6 mostly fit those of a rural village. But around noon, it has no rural quality (unless we classify agricultural machinery as rural). It is also informative about the deviations from the norm. The example just before midnight shows the high audibility of passing cars (leading to chaotic evaluations). This may be a source of annoyance because people typically use this hour to fall asleep. It also makes it possible to objectively compare different locations directly regarding the audibility of people, birds, cars, or other stationary contributions. This can be used to identify high-quality soundscapes (for which noise regulations provide no protection) and can be used by governments to decide on the priority of noise complaints.

### 4.5. Estimating Annoyance Likelihood

While working with or for highly annoyed individuals, a common question is to determine the cause and severity of an annoying source and to predict or measure the effect of noise mitigation efforts. Separation in different perceptual layers seems the key to separating the annoying sounds and estimating their audibility.

A series of experiments ([5]; [11]) investigated the detection and recognition of thresholds of environmental sounds in broadband noise. The local SNR (effectively audibility) was used to directly compare cochleograms. A local SNR of 0 dB occurred when the most audible target sound component had the same energy as the environmental sound. Known targets could be detected up to −5 dB local SNR. Unknown targets could be recognized while effortfully attending above 5 dB local SNR. Unattended, these targets became more and more salient the higher the SNR. Above 20 dB, the targets are very easily audible (attended or not) and can become dominant when they draw more attention than desired objects of attention.

A recent project performed a pre- and post-measurement test ([31]). A Dutch village was exposed to a complex harmonic source a few hundred meters away, which led to citizen complaints. These eventually led to structural changes and a new situation in which the loudness of the annoying source was reduced and its tonal character much diminished. The citizens reported often in terms of audibility. Which could consistently be coupled to a distance to a (more) stationary background. The long-term measurement allowed us to estimate audibility levels as a fraction of time ([11]). This is depicted in Figure 8.

## 5. Reflections and Conclusions

This paper outlined a way to derive, conceptualize, understand, and model the main features of sound appraisal. It approached the sound appraisal process from first principles via enactive cognition and core cognition. This led to a quadrant structure that we coupled to (1) the requirements of survival and thriving, (2) four forms of attention, (3) moods and core affect, (4) the estimation of positive indicators of audible safety, (5) the soundscape dimensions (as in the ISO standard), (6) the separation of frequency domain data into four layers, (7) modeling of sound appraisal, and (8) the sonic climate resulting from the aggregation of appraisal values. One of the *falsifiable* consequences of our approach is that we predict that all animals with a developed hearing and listening function will reflect these eight aspects.

A more direct validation of this sound appraisal methodology is needed. Currently, it complies broadly with soundscape theory derived from many experiments and, in situ, soundwalks. In addition, we reproduce the general structure of human sound appraisal experiments in which participants had to indicate pleasantness and eventfulness with joysticks ([1]; [88]). However, we need more careful experiments to fine-tune our model to match detailed human performance. This will likely lead to better-supported time-constant choices and probably more flexible or complex appraisal strategies. Yet having a starting point to falsify or improve on is useful in this phase.

This paper showed the implemented bare bones of a processing chain from a cochlea model to aggregated appraisals. The blue concepts in Figure 2 provide an idea of the type of outcomes to be produced by future model versions. Some of them are rather simple, such as the distinction between rural and urban. Others are more difficult, such as the opposition between harmonious and disharmonious, or inviting versus repulsive. Modeling the human performance of these concepts is one of the more tractable ways to understand the complex and convoluted pre-attentive processing involved in hearing and sound appraisal. In general, the type of meaning that the hearing system attributes will tell us more about the functions of audition and the ways our brainstem and other subcortical processes regulate our lives.

The modeled sound appraisal and sonic climate were applied in several projects, and it was found to be an insightful way to communicate soundscape quality. The method is tractable, explainable, and intuitive and allows for the comparison between different times of the day and year and different locations. It provides a hard, defendable, and difficult-to-argue-with representation that can—much better than loudness-based methods—be used to measure, analyze, compare, and optimize the quality of the sonic environment. It can be applied “*To make the world sound better*”.

## Figures and Tables

**Figure 1 behavsci-15-00797-f001:**
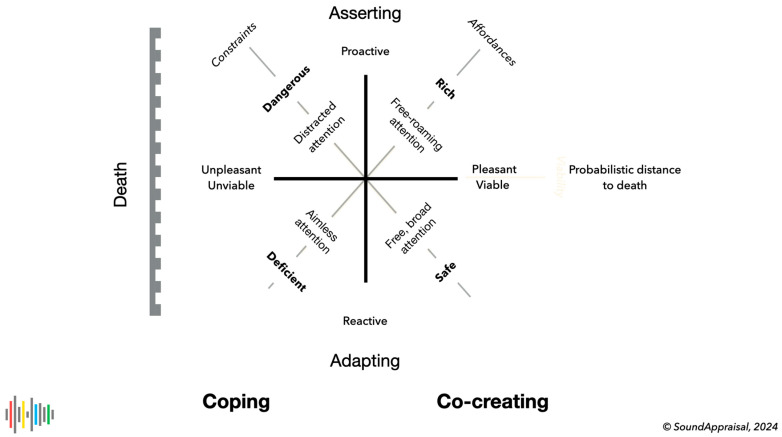
An overview of core cognition tuned to human cognition as compiled from ([14]; [7]; [8]; [90]; [36]). The horizontal axis depicts the probabilistic distance to death to be maximized. The diagonal axes depict the affordances and constraints the individual–environment relationship offers or demands. Central in the figure are associated forms of attention and, in bold, subjective evaluations of the state of the environment.

**Figure 2 behavsci-15-00797-f002:**
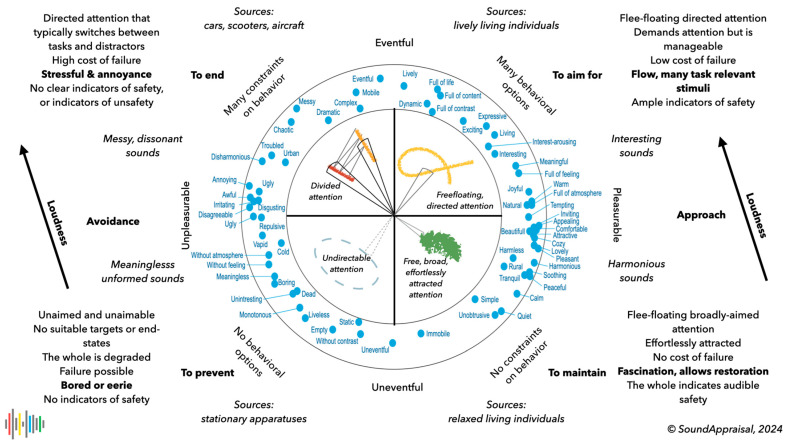
An overview of many features of hearing relevant to soundscape appraisal (Figure 1). At the center are four types of attention; the soundscape descriptors in the circle denote mood words associated with basic meaning-giving and are generally attitudes towards the (sonic) world as a whole. The words at the main axes describe the normal soundscape dimensions, while those at the diagonals refer to the dimensions of core cognition. The words in bold denote motivations. The sources indicate typical sources for each quadrant. The loudness arrows indicate the typical direction associated with raising loudness. Finally, the descriptions in the corners summarize key features of each quadrant.

**Figure 3 behavsci-15-00797-f003:**
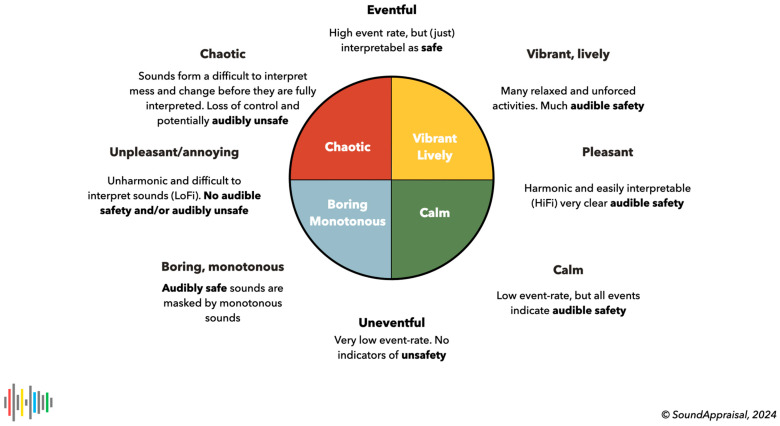
Audible safety in relation to soundscape evaluations and the quadrant structure. Note that this figure uses the soundscape terminology proposed in ISO 12913, but its formulation is based on first principles associated with estimating positive indicators of audible safety.

**Figure 4 behavsci-15-00797-f004:**
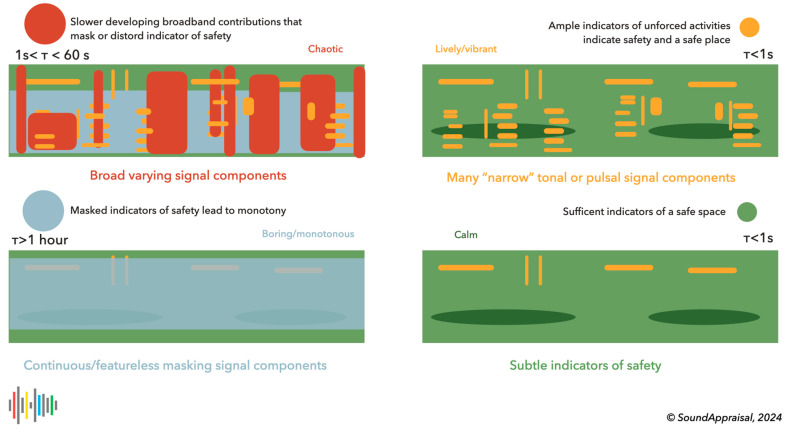
A cartoonish depiction of the four layers in associated colors. Green denotes a subtle background of proximal (usually subtle) indicators of safety and normalcy. Yellow denotes clear and prominent indicators of living beings engaged in unforced activities that indicate they feel safe. These contributions are typically narrow in frequency (tonal) or time (pulsal). The time constant of green and yellow contributions is lower than 1 s (typical for normal agentic behavior). The contributions in red are typically broadband noise with time constants between 1 s and 1 min. Red contributions mask and distort safety indicators, confusing and complicating the auditory scene analysis, and leading to a chaotic appraisal. Finally, prolonged broadband sounds in gray-blue have a longer time constant, and because they mask safety indicators, they lead to an impoverished, monotonous soundscape.

**Figure 5 behavsci-15-00797-f005:**
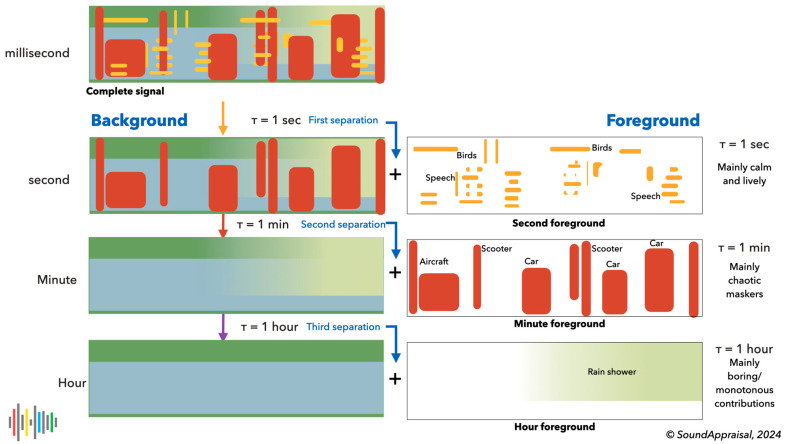
A cartoon depiction of the perceptual layer separation process. Starting from the cochleogram, signal components with a development (time constant) faster than 1 s are first isolated and subtracted. This yields the 1 s foreground that mostly indicates calmness and liveliness. Then, everything with a time constant faster than 1 min is removed to yield the minute foreground that typically masks and distorts audible safety indicators and leads to a chaotic interpretation. Finally, everything faster than 1 h is removed, here the gradual onset of a rain shower; this layer typically results in a boring or monotonous appraisal, just like the remaining hour background.

**Figure 6 behavsci-15-00797-f006:**
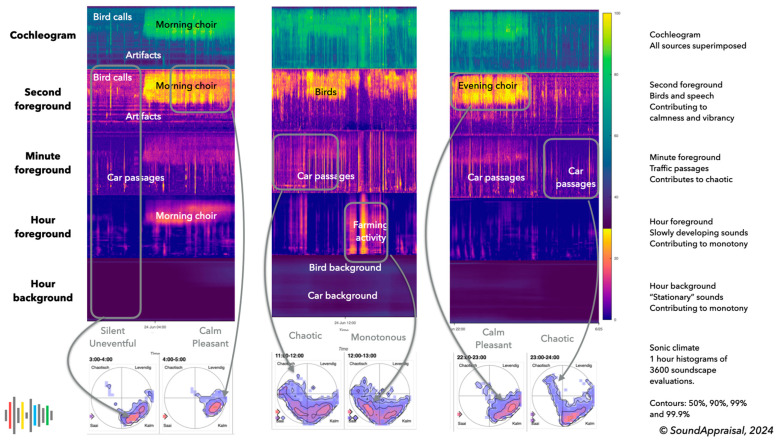
Four perceptual layer examples below the original cochleogram from a rural village on 24 June 2024. The lower circular histograms are the associated sonic climate depictions. The first panel reflects the two hours between 3 and 5 a.m.. The first hour (3–4 a.m.) is a quiet summer night with an occasional bird call and some distant car passages. The second foreground shows some horizontal artifacts that are exaggerated in this visualization (due to max pooling). At about 4 a.m., the dawn bird chorus starts, which leads to a calm and pleasant evaluation. The second panel shows 11 a.m.–1 p.m.. This contains many car passages that contribute to a chaotic evaluation. At noon, a farmer is active for about 30 min, contributing to a monotonous evaluation. The last panel shows the late evening between 10 p.m. and 12 a.m.. Initially, the evening bird choir is quite dominant. When the birds settle for the night, the village becomes mostly silent, but occasional car passages become highly audible and contribute to occasional chaotic evaluations.

**Figure 7 behavsci-15-00797-f007:**
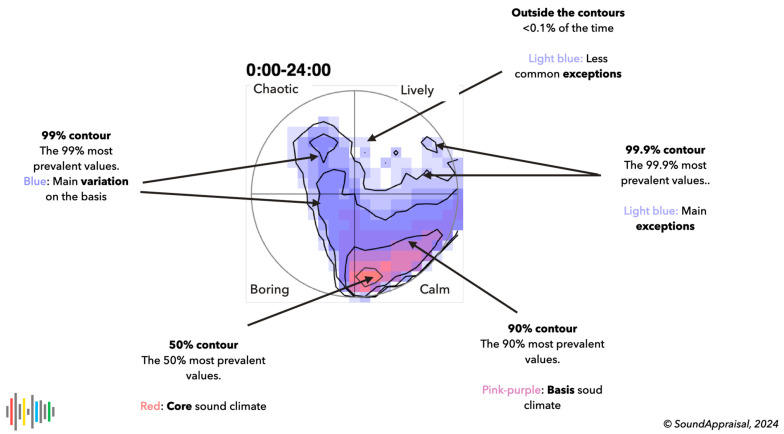
The sonic climate of a full day (86,400 s). The color coding and the contours indicate the most to least common soundscape estimations during this day. This is the typical soundscape of a quiet urban area, mostly quiet to calm. A small percentage of the time, car passages lead to a boring or chaotic soundscape evaluation.

**Figure 8 behavsci-15-00797-f008:**
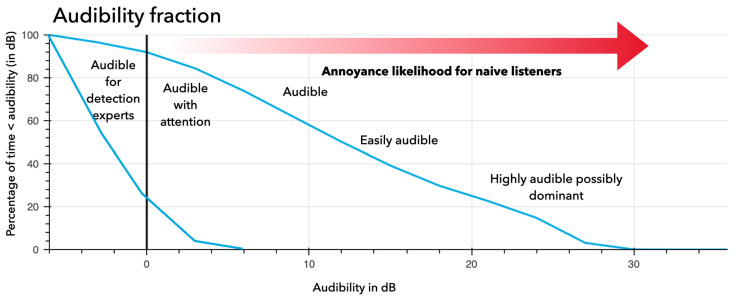
Different audibility fractions of a pre- and post-measurement test in a noise and annoyance mitigation project. Initially, the sound source had an audibility (upper blue line) higher than a 10 dB local SNR for 60% of the time. About 25% of the time, its audibility was even more than a 20 dB local SNR. Trained listeners (i.e., the local population) could estimate the source almost 90% of the time at a 0 dB local SNR. In the final situation (lower blue line), the audibility was markedly reduced to about 20% of the time for trained listeners (at 0 dB local SNR). However, still audible, it was no longer a distracting sound in the village. One villager remarked that “We have our village back”.

**Table 1 behavsci-15-00797-t001:** Example values used to compute soundscape evaluations.

Layer	Value Range	Freq. Range	Quadrant	Scope
Hour background	30–70 dB(A)	20–23,000 Hz	Boring	1 s
Hour foreground	0–20 dB	20–2300 Hz 2300–23,000 Hz	Boring Center	1 s
Minute foreground	0–15 dB	20–23,000 Hz	Chaotic	1 s
Second foreground	0–20 dB	20–2300 Hz 2300–23,000 Hz	Calm Lively	5 s

## Data Availability

No new data were created or analyzed in this study.

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
