# Peer review of "Appraising the Sonic Environment: A Conceptual Framework for Perceptual, Computational, and Cognitive Requirements"

_behavsci, 2025, doi:10.3390/bs15060797_

Round 1

Reviewer 1 Report

Comments and Suggestions for Authors

Summary: The manuscript presented a theoretical framework of how the hearing process perceive, recognize, and evaluate the auditory environment, and pass it on to higher level processing (“the listening process”). The author described the prerequisites of this process from the perspectives of cognition, physiology, acoustics, and auditory perception. The author then discussed a selection of features in the hearing process, particularly loudness-based features. Finally, the author described a model of mapping the cochleogram of environmental sounds to four perceptual layers, corresponding to different coordinates in a quadrant structure derived from the enactive cognition theory.

Overall assessments: The language quality of the manuscript is very good. However, there are quite a few concerns regarding the originality, credibility and scientific soundness of the manuscript, detailed below.

Major concerns:

(1) Most of the cited sources are literature reviews or textbooks. 33% of the cited sources are self-citations. Since the manuscript is submitted as a literature review, and that the majority of the sections either presented existing findings directly, or described theoretical frameworks heavily built upon existing findings, it is crucial to cite a variety of credible, recent and relevant first-hand empirical sources. Many contents in the manuscript are poorly supported, with only one or even no reference cited in the entire section (e.g., the “Primitive auditory scene analysis” section on page 7).

(2) Many key concepts are not clearly defined when they first appeared in the paper – the definitions usually appeared much later in the paper, or sometimes no scientific definitions were provided beside anecdotal examples. For example, the concepts of “hearing process” and “listening process” should be defined clearly (as in lines 8-9 in the abstract) at the beginning of the body of the manuscript. Replacing the table of contents (lines 67-128) with a glossary of definitions could be helpful.

(3) The manuscript should be better motivated, and its significance should be described (with sufficient support from previous empirical studies) – specifically, how is the theoretical framework similar to and different from the existing ones, and why this new framework is needed? How well do psychoacoustic and physiological results support the proposed framework? To what extent does this literature review contribute new ideas to the field of auditory science?

(4) The choice of components to (or not to) include in the framework should be better justified. For example, the “core features of hearing process” are mostly loudness-based features (and attention based on loudness). There are many other equally important (if not more important) perceptual features of sounds, such as pitch and timbre, that were not mentioned or discussed in the paper. To fully justify the structure of the manuscript, the author should not only highlight the importance of those included in the manuscript, but also sufficiently justify why other important components of auditory perception are not a part of the framework.

(5) Length distribution among topics is disproportional to their relevance to the central topic. For example, in the “enactive cognition & core cognition” section (lines 143-168), two full paragraphs were on the details of the psychology and mental health aspects of the cognition framework, which is not directly relevant to the key topics of the manuscript. In lines 262-269, a full paragraph was used to describe an anecdotal example, which could be summarized in just 1-2 sentences. The key innovation of the manuscript seems to be the computational model shown in Figure 6, yet it’s described very briefly with little technical details. I would recommend presenting the methods and results of the computational models used to derive Figure 6 in detail (if it's a novel model not yet published elsewhere), while condensing the 15 pages of background content before it into an introduction section.

(6) The manuscript contains descriptions of the author’s company not directly relevant to the topic or the context, which would be perceived as commercial advertising by the reader. Specifically, lines 29-31 stated that the manuscript "explains the research vision of our company”; the manuscript concluded with “that goes to the SoundAppraisal’s mission: ‘to make the world sound better’” (line 761); and almost the entire paragraph of lines 53-60 was dedicated to describing the company. These should be removed if the author would like to submit a revised version of the manuscript in the future.

Author Response

Dear reviewer, 

Here is a second version of the paper. Your and the other reviewers' criticism and a rereading of the paper made it clear to me that I set out to do one thing while delivering something else. This is not good. The manuscript lacked conceptual clarity, and your harsh criticism was justified and welcome.

I changed the title to match what the paper actually addresses: “Appraising the sonic environment; algorithmic requirements.” This has been a long thread in my work (in and out of academia). So, the focus now is on the algorithmic requirements for modeling sonic environment appraisal. I tried to clarify that in every part of the paper and removed some parts that did not contribute. I’m sure the paper now has just one focus and is much better motivated. Hopefully increasing its scientific soundness.

Major concerns
(1)  Most literature is from reviews or textbooks, and a third are references to self. It needs a high variety credible, recent and relevant first-hand empirical sources. And some parts of the paper need much more support.

Response: I added more than 40 references, many recent. These references support the new focus of the paper.

(2) Many key concepts, such as the hearing and listening process were not clearly defined

Response: I expanded the numbered overview in the introduction to serve both as an overview and to introduce and define key concepts. The central concept is now “sound appraisal,” which has been defined in the introduction.  

(3) Better motivation of the manuscript. “How is the theoretical framework similar to and different from the existing ones, and why is this new framework needed?” How well do psychoacoustic and physiological results support it?

Response: The focus on sound appraisal and its structure as used within soundscape research and practice is a unique and valuable contribution. The support comes in part from psychoacoustic and physiological results (now expanded in the references), but also much of the support for the algorithmic demands originates from psychology (attention, role of safety, stress, mood, emotion, psycholinguistics, speech recognition robustness, and (auditory) phenomenology).  

(4) The choice of components to (or not to) include in the framework should be better justified.

Response: This is an important remark. The explicit focus on the algorithmic demands to explain the structure of appraisal from an analysis of the signal now clearly justifies all the inclusion criteria. The clearer focus also justifies the exclusion of topics (such as directional hearing).

(5) The Length distribution among topics is disproportional to their relevance to the central topic. Mental health is not directly relevant, and a full paragraph is used for an anecdotal example. The computational model in Figure 6 needs more detail.

Response: 
The length distribution has been balanced given the focus on the algorithmic demands. The mental health aspects are removed from the paper. The anecdotal example explaining the role of source physics to be estimated from the signal to explain perceptual constancy is now clearer. The model on Figure 6 is now treated at the level of algorithmic demands and not in detail. It will be properly discussed in a future paper (hopefully) using this paper as necessary context.

(6) Unnecessary and unwelcome references to the Author’s paper.

Response:
Apart from the affiliation, all the references to the company are removed.

Reviewer 2 Report

Comments and Suggestions for Authors

The paper is interesting but requires the integration with the real auditory systems. The descriptors are a novel insight that does not correlate easily with the auditory system in humans and other mammals as well as non-mammal vertebrates. Much insight needs to have many features which are unclear and do not help to fully understand what the main takes is.  Soundscape is a first principles with sound that categorizes it into four quadrants. In the final Fig. 5 depicts the progression of separating between different speed (second, minute, hours) that yields foreground from the background and segregates from a boring/monotonous contributions.  How do we gain an insight is unclear.  Yes, proper validation is needed that is unclear how close the match with humans can become.

#43 Hearing is shared for subcortical cognition with not only mammals but other non-mammalian (birds, frogs, certain bony fishes).  Moreover, birds share with mammals with cortical region.

#46 define the primitive auditory scene?  We know that the cochlea reaches out to innervate bilateral the inferior colliculi followed the MGB to give the AC input (see a recent review by Pyott et al., 2024).

#50 Many reviews are not cited, for example the work of Grothe in a more recent book on the auditory system (B Grothe, 2020 Vol 2, 1-17)

#220 Brainstem is composed of distinct nuclei that combine are only a small part of ‘brainstem’.  Please provide a good citation for this general organization (i.e. Pyott et al., 2024 or related papers).

# 242-248  I strongly suggest building into the work of AJ King that provides the feedback from the cortex to the inferior colliculi (King, 2020, Vol 2, 732-748).  In addition, I believe the context of the amygdala is provided in Wenstrup et al 2020, 812-8370.  Specifically, certain amygdala has direct feedback to the inferior colliculi in bats but its input to mice and humans is unclear.  Therefore, the connection proposed may or may not exist in humans.

#286-293 We know that two ears are interacting to generate sound with appropriate processing to generate frequency maps and allow us to build the directionality of sound  (see Grothe, 2020,Vol2, ff)

# 295 ff I suggest to read and add the work of Shamma,2020, 777-790 and Malmierca 2020, 749-776.

Author Response

General: 
“The paper is interesting but requires the integration with the real auditory systems. The descriptors are a novel insight that does not correlate easily with the auditory system in humans and other mammals as well as non-mammal vertebrates. Much insight needs to have many features which are unclear and do not help to fully understand what the main takes is.  Soundscape is a first principles with sound that categorizes it into four quadrants. In the final Fig. 5 depicts the progression of separating between different speed (second, minute, hours) that yields foreground from the background and segregates from a boring/monotonous contributions.  How do we gain an insight is unclear.  Yes, proper validation is needed that is unclear how close the match with humans can become.”

Response:

Here is a second version of the paper. Your and the other reviewers' criticism and a rereading of the paper made it clear to me that I set out to do one thing while delivering something else. This is not good. The manuscript lacked conceptual clarity, and your harsh criticism was justified and welcome.

I changed the title to match what the paper actually addresses: “Appraising the sonic environment; algorithmic requirements.” This has been a long thread in my work (in and out of academia). So, the focus now is on the algorithmic requirements for modeling sonic environment appraisal. I tried to clarify that in every part of the paper and removed some parts that did not contribute. I’m sure the paper now has just one focus and is much better motivated.

The current manuscript relies less on “credible, recent, and relevant first-hand empirical sources.” Instead, it depends more on several computational requirements that have been around for a long time but that, in most cases, have barely been made explicit or that address only a few aspects in isolation. I aim to provide a comprehensive and step-by-step logical overview explaining why sound appraisal has its structure and what types of processing are necessary to model it.

#43 Hearing is shared for subcortical cognition with not only mammals but other non-mammalian (birds, frogs, certain bony fishes).  Moreover, birds share with mammals with cortical region.

Response: Agreed. However, since the focus is (now) on algorithmic requirements, the link with how different species have satisfied these is not a central focus. Science indicating that they satisfy different requirements is, of course, highly relevant.

#46 define the primitive auditory scene?  We know that the cochlea reaches out to innervate bilateral the inferior colliculi followed the MGB to give the AC input (see a recent review by Pyott et al., 2024).

Response: I’ve added the relevant reference.

#50 Many reviews are not cited, for example the work of Grothe in a more recent book on the auditory system (B Grothe, 2020 Vol 2, 1-17)

Response: Including this one, I have added 40 references from a wide range of contributing domains mainly from diverse brances of psychology:  (attention, role of safety, stress, mood, emotion), but also speech recognition robustness, psycholinguistics, and (auditory) phenomenology.

#220 Brainstem is composed of distinct nuclei that combine are only a small part of ‘brainstem’.  Please provide a good citation for this general organization (i.e. Pyott et al., 2024 or related papers).

Response:
Done

#242 - 248  I strongly suggest building into the work of AJ King that provides the feedback from the cortex to the inferior colliculi (King, 2020, Vol 2, 732-748).  In addition, I believe the context of the amygdala is provided in Wenstrup et al 2020, 812-8370.  Specifically, certain amygdala has direct feedback to the inferior colliculi in bats but its input to mice and humans is unclear.  Therefore, the connection proposed may or may not exist in humans.

Response:
Given the focus of sound appraisal, I focus less on the difference brainstem nuclei, and more on appraisal, hence I have added ample references to Porges (e.g., Porges, S.W. Polyvagal Theory: A Science of Safety. _Front. Integr. Neurosci._ **2022**, _16_). He has a very useful concept to describe estimating safety: Neuroception

#286 -293 We know that two ears are interacting to generate sound with appropriate processing to generate frequency maps and allow us to build the directionality of sound  (see Grothe, 2020,Vol2, ff)

Response:
I had decided to exclude directional hearing.

#295 ff I suggest to read and add the work of Shamma,2020, 777-790 and Malmierca 2020, 749-776.

Response:
I have added Shamma’s work as a reference. Instead of Malmierca, I added references to Bendixen because it fitted more in the flow of the paper.

Round 2

Reviewer 1 Report

Comments and Suggestions for Authors

Thank you for the opportunity to review the revised manuscript. The manuscript has been significantly improved compared to the previous version. My comments, detailed below, are mostly on the clarity and soundness of the content.

Page 1, title: the title is a bit fragmented and does not effectively convey the core idea of the paper. There’s a sentence in lines 44-45: “This paper aims to provide a first-principles derivation of the perceptual and cognitive processes underlying soundscape appraisal and their computational requirements.” This sentence summarizes the key content of the paper very well and could be rephrased into a title. A possible title would be “Appraising the sonic environment: A conceptual framework for the algorithmic requirements of the underlying perceptual and cognitive processes” – please feel free to revise as needed to better express the main idea of the paper.

Page 1, overall: It might be worth emphasizing in the title, the abstract, and the introduction that the paper presents a conceptual and theoretical framework, rather than the algorithmic requirements of a specific mathematical model of the sound appraisal process.

Page 1, lines 7-8: “Sound appraisal leads to auditory sense-making and the soundscape as the perceived and understood acoustic environment.”  - This sentence is unclear. Do you mean “… the soundscape is the perceived and understood acoustic environment”, or “in the process of auditory sense-making, the soundscape is interpreted as the perceived and understood environment”?

Page 1, line 17: “Applied, …” – It is unclear what is being applied to what here.

Page 2, line 56: “attention research” appeared twice in this list.

Pages 2-3, lines 65-144: This section has been improved compared to the previous version but still could be better supported. Please cite appropriate references to support these definitions and basic concepts. 

Page 3, lines 148-150: “The hearing process can only be understood in the wider context of the cognitive process …” This might be an overly strong statement. There are various possible ways to understand the hearing process, not just through the cognitive process of survival and flourishing.

Page 6, lines 236-251: The attentive process described in this paragraph is a specific form of attention, commonly referred to as the “bottom-up” or stimulus-driven attention. This could be added to the description to avoid potential overgeneralization, and also to better distinguish from the knowledge-driven (top-down) attention described in the next section.

Page 7, lines 330-332: This sentence is unclear. Do you mean “the characteristic temporal development of sources can be used to separate perceptual layers”? Also, since the discussion only included the hearing process in this section, it could be added how the ASA is achieved in the listening process, and how the hearing and listening processes work together in the process of ASA.

Page 8, section 3: Although titled “core features of the hearing process”, this section heavily focused on loudness perception (including audibility and noise sensitivity). It could be added why the other psychoacoustic properties (pitch and timbre) are not included in this section as a “core feature”.

Page 10, liens 429-430: “This allows processing a high event rate” – unclear. Does this sentence mean that the HiFi soundscape allows processing with a high fidelity, processing a larger number of events at the same time, or at a high speed?

Page 11, lines 459-467: The tendency of the preattentive cognitive tends to “err on the side of safety” should be supported with experimental evidence and references, on top of anecdotal examples.

Page 12, lines 537-543: The definition of “divided attention” here is different from what it generally refers to in cognitive psychology. In psychology, the term generally refers to the ability to focus on multiple tasks at the same time, rather than focusing on one task while being easily distracted by noises. This difference might be worth further clarifying.

Page 13, lines 552-562: Were the time scales of the different types of sounds obtained from previous research or common sense/anecdotal observations? The conclusions in lines 563-567 are fairly strong and should be supported by more convincing evidence. Also, using the time scale to separate animal- vs. industrial-sourced sounds may not be a very reliable clue, as the animal sounds could be on a scale of 5-30 seconds or even minutes, and the industrial/artificial noises could be as short as 1 second.

Page 17, Figure 7: The figure is interesting, but it’s not discussed in the text. How was the figure acquired? Were the coordinates of the grids determined by a computational model, or perceptual categorization of the listeners?

Page 18, Figure 8: Similarly, more information could be provided about how this figure was derived. The project described in lines 727-731 doesn’t seem to be in the reference list so providing more details of the research project would be helpful.

Author Response

Thank you again for some excellent and highly constructive remarks. 

Page 1: I changed the title to: “Appraising the sonic environment: a conceptual framework for perceptual, computational, and cognitive requirements”

page 1, 7-8: Improved the sentence: “Sound appraisal involves auditory sense-making and produces the soundscape as the perceived and understood acoustic environment.”

page 1: Applied → Combined (the previous)

page 2: 2nd attention research removed

page 2-3: Added relevant references to overview list 

page 3: “Can only be understood” → “Can be understood”

page 7: addition of common attention descriptors → “Unexpected, context-inappropriate, or simply loud sounds can lead to a brainstem response – a form of bottom-up or stimulus-driven attention – in which ongoing neocortical processes are interfered with to address sensations that the brainstem considers worthy of attentive processing [26,29,50,51]. Attentive – top-down or knowledge-driven – processing requires resources that become available through arousal.”

page 7: Unclear sentence  → "In our modeling, we use the characteristic temporal development of sources to separate perceptual layers (using the old-plus-new heuristic), implementing primitive auditory scene analysis. It seems that we do not need schema-based or knowledge-driven ASA to explain the main features of sound appraisal. Hence, we conclude that sound appraisal is an outcome of the hearing process."

page 8: Section 3: Why not address pitch and timbre → Although pitch, timbre, roughness, and sharpness are clearly appraisals and outcomes of primitive auditory scene analysis. However, we ignore these here since the practice of soundscape and noise abatement focuses (too heavily) on loudness. In addition, these involve additional complexity extending beyond the current scope of the paper.’ 

As a side note: we have just implemented a “heavy-duty” harmonic complex estimation algorithm that is designed to break-down gracefully (and informatively) as the local SNR (or signal-to-target ratios) declines. We have not yet had any true experience with it. 

Page 10: Unclarity about processing a high event rate → ”A HiFi soundscape has few overlapping sources and well-audible signal components that primitive auditory scene analysis processes can easily estimate and analyze. Hence, these can recruit the top-down knowledge necessary to categorize and attribute meaning. This allows processing at a higher event rate than a LoFi environment allows. In a LoFi environment, the signal often changes before it is fully processed, contributing to a lack of certainty about the state of the environment and low audible safety.”

page 11: Basis of preattentive cognition as “to err on the side of safety” → This is a basic evolutionary argument for which I have only indirect evidence. I changed it to “From an evolutionary perspective, estimating audible safety must be fast and “better safe than sorry.” Hence, it errs on the side of safety. Preattentive cognition must keep the animal safe in a complex world, and it can only do that with basic learning processes such as conditioned responses and habituation.”

Page 12: divided attention. → Indeed, divided attention is not the proper term here. So I introduced the term ‘distracted attention’. 
_Distracted attention_ can be defined as a situation when you aim to focus on a task, but you are distracted by, for example, disturbing noises that prevent you from concentrating or force you to switch between a self-selected task and attending to a distractor. This requires a higher level of arousal [36] and is an inefficient use of time and energy. Not only because you focus on an unwanted stimulus that likely elicits negative emotions, but also because it requires considerable switching time to return to your task [85]. This is (rightly) perceived as highly disturbing, tiring, and annoying [77].

Page 13: Conclusion is too strong. → I changed “conclusion” to “proposal”. 

Page 17: Figure 8 not discussed →  Number changed to Figure 7 And I added more precision and information to yield: 
“The sonic climate is the long-term aggregate of soundscape evaluations as a 2D histogram. Ideally, it approaches the effort of the average outcome of human volunteers who produce a second-by-second response to the ISO standard question. Figure 7 shows what results from binning all pleasantness and eventfulness combinations in a 2D histogram (here 21x21 for all seconds of a day) on axes spanning [-1, 1]. Figure 7 shows indications for representing, color coding, and interpreting a sonic climate histogram. The color coding scales with the fraction of the data per bin. Currently, no claims are made about the correspondence of the automatic and the human appraisal estimation other than that the resulting 2D roughly corresponds to expectations and is informative even for laymen.”

Page 18: More information on Figure 8 → The project was for the Dutch Ministry of Defense and the report is in Dutch. It is not classified, but not (yet) intended for broader dissemination. I changed the text to: 
“A recent project performed a pre- and post-measurement [90]. A Dutch village was exposed to a complex harmonic source a few hundred meters away, which led to citizen complaints. This eventually led to structural changes and a new situation in which the loudness of the annoying source was reduced and its tonal character much diminished. The citizens reported often in terms of audibility (Dutch: “hoorbaarheid”) that could be easily and consistently coupled to a distance to a (more) stationary background. The long-term measurement allowed us to estimate audibility levels as a fraction of the time [89]. This is depicted in Figure 8.”

[90] Cobussen, M.; Andringa, T.C. _Een Geluidsbelevingsonderzoek Naar de Radarpost in Wier_; Sound Studies Center, Universiteit Leiden, 2024; p. 62;.

Apart from addressing the reviewers’ comments, I addressed some consistency issues and added a few more references. 

Finally, I highly appreciate your detailed and constructive comments. I have reflected this in the Acknowledgments. 

Reviewer 2 Report

Comments and Suggestions for Authors

The paper is improved but has mior changes are needed.

#31 It does not make sense to start with Fig. 2+3 (#31) but should start with Fig. 1.  Alternatively, kill the Fig. 2+3, you use it later.

#52 or #57  I suggest to expand the early hearing development and the loss of older people.  Please take a look at the two papers and can expand it (PMID: 37626546; PMID: 35250542).    

Author Response

Thank you for your comments. One important change is the title, it is now: Appraising the sonic environment: a conceptual framework for perceptual, computational, and cognitive requirements”, because this describes the content better. 

L 31: The figures are positioned where they are explained. In principle, I can even omit the references, although they are illustrative. As a compromise, I have put the reference between brackets. Figure 1 does not make a sound appraisal point. 

L52&57: I could not incorporate early hearing development and age-related degradation as a topic without disrupting the flow of the paper. 

Round 3

Reviewer 1 Report

Comments and Suggestions for Authors

Thank you for the opportunity to review this manuscript. My comments on the last version have been sufficiently addressed, and I recommend the manuscript for publication. There are only a couple suggestions: (1) Page 6, line 243: this section is more in line with the concept of bottom-up attention, rather than top-down attention. It has been specifically stated on page 8 that “we do not need schema-based or knowledge-driven ASA to explain the main features of sound appraisal”. Therefore it might work better if the discussion on page 6 is limited to bottom-up attention (and either remove the sentence about top-down attention, or make a distinction between these two types of attention before focusing on bottom-up attention). (2) Figure 8: Both lines are in blue, so it might be confusing that the figure caption used color to direct the reader to one of the lines “initially, the sound source had an audibility (blue line) higher than 10dB local SNR…”. I recommend either changing the color of the line for the final situation, or replacing “blue line” with something like “the upper line” in the figure caption. The author could decide whether or not to implement these changes in the manuscript. Either way, the manuscript is in good shape.